# Adjuvant Radiation Therapy in Desmoplastic Melanoma: A Scoping Review

**DOI:** 10.3390/cancers16223874

**Published:** 2024-11-19

**Authors:** Christina Setareh Sharafi, B. Ashleigh Guadagnolo, Kelly C. Nelson, Devarati Mitra

**Affiliations:** 1College of Osteopathic Medicine, NOVA Southeastern University, Fort Lauderdale, FL 33328, USA; setarehsharafi96@gmail.com; 2Department of Radiation Oncology, The University of Texas MD Anderson Cancer Center, Houston, TX 77030, USA; aguadagn@mdanderson.org; 3Department of Dermatology, The University of Texas MD Anderson Cancer Center, Houston, TX 77030, USA; kcnelson1@mdanderson.org

**Keywords:** desmoplastic melanoma, radiation therapy, local control, immunotherapy

## Abstract

Desmoplastic melanoma (DM) is a rare subtype of cutaneous melanoma with a distinct clinical presentation, natural history, and response to standard therapies. This review evaluates the role of radiation therapy (RT) in the management of DM, in the context of evolving surgical recommendations and systemic therapy options.

## 1. Introduction

Desmoplastic melanoma (DM) is a rare histologic subtype of melanoma, accounting for 1–4% of cutaneous melanomas [1,2,3,4]. In 1971, Conley and colleagues defined DM as malignant spindle cells separated by prominent fibrocollagenous stroma [5]. Risk factors for DM mirror those for cutaneous melanoma broadly and include male gender, older age, and chronic sun exposure [1,2,4,6].

Like other forms of cutaneous melanoma, the incidence of DM has been increasing over the past 15 years [6]. In parallel, new management strategies have emerged, including evolving perspectives on sentinel lymph node biopsy, immunotherapy, and radiotherapy (RT). This review seeks to explore the role of adjuvant RT in the contemporary management of DM.

## 2. Pathology and Histology

DM can be challenging to diagnose. Primary tumors can be either amelanotic or melanotic, often resembling other cutaneous malignancies or even benign scar tissue or dermatofibroma [7,8]. Histologically, DM presents as spindle-shaped melanocytes embedded in a dense, fibrous (desmoplastic) collagen matrix, with diminished cellularity relative to other cutaneous melanomas [1]. In this context, DM primary tumors typically present with a high T-stage and an elevated Breslow thickness [3]. The diagnosis of DM is also challenging due to its tendency to test negative for traditional melanoma markers, such as Melan-A and HMB-45. In addition, while individual DM cells are typically S100 positive, this characteristic is not always helpful, as much of the tumor bulk consists of stromal tissue [1,2].

In the literature, DM is often lumped together with neurotropic melanoma; however, these are distinct entities. While DM frequently involves nerve fibers (making it appear neurotropic), not all cases of DM are neurotropic, and not all neurotropic melanomas are classified as DM. Of note, a cutaneous melanoma of any subtype can present with perineural invasion, which is an independent risk factor for local recurrence [7].

In 2004, two subtypes of DM were distinguished based on histologic features: (1) pure desmoplastic melanoma (pDM) and (2) mixed desmoplastic melanoma (mDM). pDM is defined as a tumor consisting of more than 90% stromal fibrosis, with a paucicellular distribution of spindle-shaped melanocytes. In contrast, mDM is defined as a tumor containing 10–90% densely cellular nonfibrotic areas with higher rates of mitoses [5,8,9]. The emerging consensus indicates that mDM and pDM are two distinct entities, with mDM behaving more like other subtypes of cutaneous melanoma. As such, mDM is more likely to demonstrate lymphovascular invasion and nodal involvement, along with a higher proliferation rate as assessed by Ki-67 [9,10]. Despite clear clinical differences, the underlying biological mechanisms distinguishing these two subtypes remain unknown.

## 3. Surgical Management

The primary treatment for localized DM, as with any melanoma, is wide local excision [11]. These lesions commonly present with >2 mm Breslow thickness and thus standard 2 cm resection margins are typically recommended [3,12].

Localized pDM has an elevated risk of local recurrence with nodal progression being less common. While evidence supports Sentinel Lymph Node Biopsy (SLNB) for most thick cutaneous melanomas, there is controversy regarding whether pDM requires SLNB [13]. mDM appears to behave like most cutaneous melanomas such that patients with risk factors for nodal disease (e.g., >1 mm thick primary tumors) benefit from SLNB [14,15]. However, pDM primary tumors are less likely to exhibit lymphovascular invasion and lower rates of sentinel lymph node (SLN) positivity [13]. Several studies highlight the rarity of SLN involvement in pDM. In a cohort study of 244 head and neck DM patients undergoing SLNB, only 9 SLN+ patients (3.7%) were identified [16]. Another study involving 250 patients with DM across various anatomical sites found that, among the 123 pDM and 129 mDM cases, only 17 (6.8%) patients had positive SLNs [5]. In the context of a small number of SLN+ patients, there was no significant difference in SLN+ rate between pDM and mDM. In a study by Laeijendecker et al., 239 patients were assessed, with 114 pDM and 123 mDM cases. Among the 62 patients who underwent SLNB, only 6 had positive SLNs, all of which were in the mDM group. Mohebati et al. studied 47 patients undergoing wide local excision for primary DM, of which 21 had SLNB, none of which harbored SLN+ disease. Taken together these studies suggest pDM has a very low SLN+ rate and may not routinely require SLNB.

## 4. Systemic Therapy

Immune checkpoint inhibition has revolutionized the management of melanoma broadly but for DM this is particularly evident. In 2018, investigators evaluating pathology reports from >1000 advanced melanoma patients treated with anti-PD1/PD-L1 immune checkpoint inhibitors found that 70% of patients with DM had an objective tumor response [17]. This led to the Southwest Oncology Group (SWOG) S1512 trial which enrolled exclusively DM patients. The first cohort from this study received neoadjuvant intent pembrolizumab and reported a 55% complete pathologic response rate. A second cohort of DM patients with unresectable disease who received pembrolizumab were found to have a 90% RECIST overall response rate with 33% complete response [18].

For patients with less advanced disease, KEYNOTE-716 found that cutaneous melanoma patients with Stage IIB and IIC disease (agnostic to histologic subtype) have significantly longer recurrence free survival with adjuvant pembrolizumab vs. placebo (1-year RFS 90% vs. 83%; at second interim analysis: HR 0.61, 95% CI 0.45–0.82) [19]. While the authors did not report the proportion with pDM, given the typically thicker Breslow thickness of this histology and prior data regarding efficacy of anti-PD1 for pDM, we would expect a similar RFS benefit in this cohort as well [17,18].

## 5. Radiation Therapy

To evaluate the role of RT in the management of resected DM we undertook a literature review. Starting with a PubMed query of “(desmoplastic melanoma) AND (radiation) OR (radiotherapy) OR (radiation therapy))”, 85 records were identified. The PRISMA diagram in Figure 1 illustrates how this search was further narrowed. Specifically, after reviewing titles for relevance to the management of DM, 41 papers were identified as on topic. Abstracts were then reviewed to exclude case reports, commentaries or reviews. The full-length manuscript was then evaluated for 21 reports with only those that discussed resected DM, RT and/or local control being included in the final review.

Overall, 10 studies were assessed as most relevant to an evaluation of the role of RT in the management of resected DM (Table 1).

Given DM’s propensity for local recurrence after surgery alone, there has been long-standing interest in adjuvant therapy options to maximize local control. This interest is particularly relevant in cosmetically sensitive areas, such as the head and neck, where wide margins are difficult to achieve due to adjacent critical anatomy [4,29].

In the early 2000s, adjuvant RT for primary DM began to be formally evaluated. An early report from UCLA included 21 patients with DM who underwent surgical resection between 1976 and 1997 [28]. In this study, 14 (67%) received adjuvant RT with a median dose of 50 Gy, administered in 1.8–2 Gy/fractions, while 7 (33%) were managed with observation alone. Patients receiving adjuvant RT experienced 100% local control. Meanwhile, 4 of the 7 (57%) patients in the surgery-only group had a local recurrence.

A 2008 study from Australia reported outcomes for 24 DM patients treated between 1997 and 2006 with excision followed by adjuvant RT [27]. In-field local recurrence was seen in only 2 patients, with both having surgical margins < 10 mm and recurrence events occurring at the field margin.

Another 2008 study conducted in Sydney identified 128 patients diagnosed with DM from 1996 to 2007 [26]. After excision, 27 patients (21%) received adjuvant RT. Patients treated in the earlier era (1996–2000) received 33 Gy in 6 fractions, administered twice weekly, whereas patients treated after 2000 received doses of 2–2.4 Gy per day. There were a total of 8 local recurrences with 6 occurring in patients from the cohort having surgery alone, and 2 local recurrences in patients receiving adjuvant RT. Although this difference was not statistically significant, given the relative imbalance in the number of patients receiving RT and likely selection bias, the authors appropriately concluded that further studies were needed.

In the same 2014 issue of the journal, Cancer, MD Anderson Cancer Center and Moffitt Cancer Center each published their respective institutional experiences treating DM with either surgery alone or surgery followed by adjuvant RT. The MD Anderson experience evaluated patients treated between 1985–2009, with 59 (45%) receiving surgery alone and 71 (55%) receiving adjuvant RT [11]. With a median follow-up of 6.6 years, 24% (n = 14) of patients receiving surgery alone experienced local recurrence while 7% (n = 5) of patients receiving adjuvant RT similarly recurred. This translated to a 10-year local control (LC) rate of 74% vs. 91% with RT (*p* = 0.009), with local control defined as absence of recurrence within 2 cm of the resection bed. This difference was driven by the cohort with pDM (46 of whom received surgery alone and 54 of whom received adjuvant RT). Specifically, 10-year LC for the pDM subset was 70% with surgery alone vs. 92% with adjuvant RT (*p* = 0.002). In contrast, the cohort with mDM (13 of whom received surgery alone and 17 of whom received adjuvant RT) had no difference in 10-year LC (88% for both). The Moffitt experience evaluated patients treated 1989–2010 with 164 (59%) receiving surgery alone and 113 (41%) receiving adjuvant RT [25]. With a median follow-up of 43.1 months, 17% (n = 28) of patients receiving surgery alone experienced local recurrence while 7% (n = 8) of patients receiving adjuvant RT similarly recurred. This translated to a 5-year LC rate of 76% vs. 95% with RT (*p* = 0.015). Unlike the MD Anderson study there did not appear to be a difference in local control related to histology being pDM vs. mDM but this may have been confounded by more pDM patients receiving adjuvant RT. At both centers, adjuvant RT was most commonly delivered as 30 Gy in 5 fractions prescribed to the point of maximum dose (D_max_), delivered twice weekly with either electrons, photons, or a combination of both. In other words, the majority of the target volume of interest received 90% of the prescribed dose, or 27 Gy, while only point doses of 30 Gy were allowed, as per the original method of prescribing this regimen devised by Ang and colleagues [30].

In 2015, a subsequent study of 316 patients with DM treated at Mofitt between 1993–2011 focused on the prognosis of the 55 patients with nodal disease [24]. While primary site RT and local control were not the focus of this report, 114 patients (36%) received adjuvant RT to the primary resection site. Outcomes were not stratified based on the application of adjuvant RT; however, 30 patients (9.5%) from the overall group experienced an isolated local recurrence.

Emory Winship Cancer Institute reported their experience with 95 DM patients in 2016 [23]. This cohort was treated between 2000 to 2014 with only 10 patients (12%) receiving adjuvant RT and the remaining 85 patients (89%) undergoing surgery alone. Despite this imbalance, 10 patients (11.8%) receiving no adjuvant therapy developed a local recurrence while none of the patients receiving adjuvant RT similarly recurred.

NCCTG N0275 was subsequently published as a single-arm prospective study of 20 DM patients treated with adjuvant 30 Gy in 5-fractions [22]. This cohort was accrued 2003–2009 across 7 sites and specifically required central pathology review to ensure only those with “predominant histologic pattern” DM were enrolled and patients with non-desmoplastic neurotropic melanoma were excluded. All patients received the prescription of 30 Gy in 5 fractions to D_max_ delivered twice weekly. With a median follow-up of 52 months, 2 patients developed local recurrence (both with surgical margins < 2 cm). This translated to a 5-year LC of 90%. There were no grade 3 toxicities. Acute radiation dermatitis (RD) was common with 15 patients (75%) experiencing grade 1 RD and 3 patients (15%) experiencing grade 2 RD.

Typically, the primary benefit of adjuvant RT is thought to be in-field disease control. However, a hypothesis-generating NCDB analysis of 2390 localized DM patients treated 2004–2013 with surgery alone (n = 2082) or adjuvant RT (n = 308) showed a potential overall survival benefit to adjuvant RT [31]. This was specifically seen with multivariate analysis (HR 0.75, 95% CI 0.58–0.97, *p* = 0.03) as well as a propensity score-matched analysis of 267 patients receiving surgery alone and 267 patients receiving adjuvant RT (5-year OS 66.5% vs. 71.8%, *p* = 0.034).

Several more recent studies have evaluated the role of adjuvant RT in a combined cohort of neurotropic melanoma patients including a subset with DM. The first of these was a 2017 a study from Australia evaluating clinicopathologic features, management and outcomes for 671 patients treated between 1985–2013 [21]. 480 patients (72%) were reported as having DM. Only 82 (12%) in the overall cohort received adjuvant RT to the primary site (with a median dose of 48 Gy in 20 fractions). Multivariable analysis illustrated adjuvant RT was able to significantly reduce the likelihood of local recurrence with HR 0.30 (95% CI 0.13–0.69, *p* = 0.005).

The recently published RTN2 Trial 01.09 serves as the only randomized study of adjuvant RT including a significant proportion of DM patients. However, patients enrolled on this study were defined as having cutaneous neurotropic melanoma of the head and neck, with only a subset having DM [20]. This study was initially designed with the goal of enrolling 100 patients and accrued from 2009–2020 but was closed prematurely due to the slow enrollment. Ultimately 50 patients were enrolled and randomized after surgical resection to observation (n = 23) vs. adjuvant RT (n = 27) delivered as 48 Gy in 20 daily fractions. Notably, the authors reported the subset of patients who had desmoplasia present but do not specifically annotate whether these represented pDM or mDM. “Overall desmoplasia” was observed in 69% of the total cohort with 13 (56.5%) in the observation arm and 21 (80.8%) in the adjuvant RT arm, suggesting an imbalance between cohorts. Data from 23 patients randomized to observation and 24 patients randomized to RT were available for the final analysis. At a median of 4.8 years follow-up, local recurrence was reported for 3 patients in the observation group and 1 patient in the adjuvant RT group. Due to the low accrual, this study was inadequately powered to establish whether adjuvant RT provided a local recurrence benefit.

## 6. Future Questions and Conclusions

To date, the RTN2 study is the only randomized study evaluating the role of RT in a cohort of patients that was enriched for DM. However, there is a lack of randomized data specific to pDM, and given the relative rarity of this tumor, it is unlikely that such data will become feasible to obtain. Therefore, we must rely on existing large retrospective cohort data for guidance regarding optimal local therapy approaches. The addition of RT to surgical resection for DM does not appear to adversely affect the toxicity profile of local disease management (no grade 3 toxicities were reported in NCCTG N0275, and 2.5% greater grade 3 toxicity in RTN2). This contrasts with adjuvant single-agent anti-PD1 therapy, which is generally regarded as well-tolerated, yet demonstrated a 9% increase in grade 3 toxicities compared to placebo in KEYNOTE-716 [19]. Thus, given the >50% reduction in local recurrence risk achieved with adjuvant RT for pDM, we maintain that adjuvant RT should continue to be recommended for this histology [11,25].

While the primary focus of this review has been evaluating the optimal adjuvant therapy for resected primary DM, recent evidence also supports the valuable role of RT in managing advanced melanoma in the contemporary era. A recent multi-institutional evaluation of 98 patients who underwent lymphadenectomy between 2010–2019 found a numerically lower rate of nodal recurrence in patients receiving adjuvant nodal RT vs. surgery alone (13.9% vs. 25.2%), though this difference was not statistically significant [32]. Additionally, results from the “Radvax” trial, presented at ASTRO, suggest that combining hypofractionated RT and pembrolizumab may yield an even greater effect in patients who are progressing on anti-PD1 therapy. Remarkably, 7 out of 16 patients experienced abscopal responses, an unprecedented outcome in this field [33]. We eagerly await the final publication and are particularly interested in whether melanoma subtype may influence the response rate to combination therapy.

For pDM specifically, a recent case report from 2023 described the outcomes of two DM patients with extensive primary tumors of the nasal dorsum and ala who declined rhinectomy. Instead, they received anti-PD1—pembrolizumab for one patient and nivolumab for the other [34]. Clinical response was partial in one case and temporary in the other. Subsequently, both received definitive RT (60 Gy in 30 fractions), resulting in the expected grade 2 radiation dermatitis. However, notably, both patients achieved a clinical complete response and remain disease-free at 6 years and 20 months, respectively. Thus, while surgery will likely continue to be the gold standard for treating primary DM, these cases highlight that for well-selected patients for whom resection is morbid, combining anti-PD-1 and RT may be an approach worth investigating prospectively.

Much remains unknown in defining the optimal treatment for patients with DM. Given that an RTN2-like study focused only on adjuvant primary site RT for pDM is unlikely to gain traction, a potential novel area for investigation could be the role of RT in the management of pDM patients with metastatic disease. Specifically, prospective evaluation of hypofractionated RT with anti-PD1 therapy in the context of progression after immune checkpoint inhibition (similar to Radvax), could be an exciting proposition.

Despite evolving recommendations for surgery and systemic therapy in DM, there is no robust published evidence to suggest that the role for adjuvant RT has diminished. Given several decades of studies that have shown adjuvant RT has a significant, lasting positive impact on local control for DM patients with limited toxicity, we contend that this therapeutic approach continues to provide significant value in disease control without significantly compromising patients’ quality of life.

## Figures and Tables

**Figure 1 cancers-16-03874-f001:**
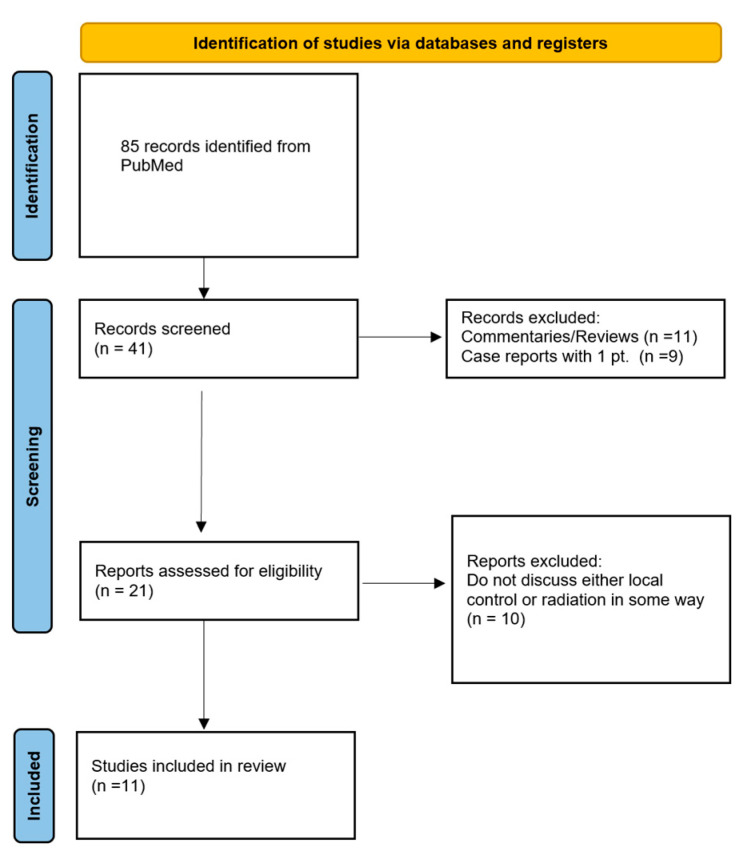
PRISMA flow diagram detailing the database searched, the number of records screened, and the studies selected for inclusion in the literature review of adjuvant radiation therapy in desmoplastic melanoma.

**Table 1 cancers-16-03874-t001:** Summary table of studies included in this review.

Author Year, Journal (Study Type)	Institution	Radiation Details	Inclusion Criteria	Number	Primary Site Treatment	Follow-Up	LR After RT	LR Comparison to No RT
RTN2 Trial 01.09 Pinkham 2024, ASO [20] (prospective, randomized study)	15 international centers (primarily Australian)	48 Gy/20 fx (1.5 cm margin) electrons or photons	Neurotropic melanoma of H&N, R0 resection	Randomized: 23 obs vs. 27 postop RT	Excision +/− RT	Median 4.8 yrs	4%	No RT: 13% HR 0.29 (95% CI 0.03–2.76, *p* = 0.28)
Varey 2017, Mod Pathol [21] (retrospective)	Melanoma Institute Australia	48 Gy/20 fx Modality not reported	Resected neurotropic melanoma, 1985–2013	671 (589 obs vs. 82 postop RT)	Excision +/− RT	Median 3.5–3.6 yrs		HR 0.30 (95% CI 0.13–0.69, *p* = 0.005
NCCTG N0275, Alliance study Rule 2016, Cancer Medicine [22] (prospective, single arm)	Mayo Clinic	30 Gy/5 fx (2–3 cm margin) electrons	R0 resected DM, >1 mm depth or locally recurrent	20	Postop RT	Median 52 mo (range 30–65)	10%	
Oliver 2016, Melanoma Res [23] (retrospective)	Emory	Mixed: 30 Gy/5 fx (40%), 48 Gy/20 fx (30%), 50–66 Gy/20–33 fx (30%) 40% electrons, 60% photons	Resected DM 2000–2014	95 (85 obs vs. 10 postop RT)	Excision +/− RT	Median~3 years	0%	No RT: 12%
Han 2015 PLOS One [24] (retrospective)	Moffitt	Not reported	Resected DM 1993–2010	316 (202 obs vs. 114 primary site RT)	Excision +/− RT	Median 5.3 years		87 any recur, 30 “isolated LR” (9.4%) not annotated by RT
Guadagnolo 2014, ASO [11] (retrospective)	MD Anderson	Mostly 30 Gy/5 fx (3–4 cm margin) 73% electrons, 25% incl photons, 1% IMRT	Resected DM between 1985 and 2009	130 (59 obs, 71 postop RT)	Excision +/− RT	Median 6.6 years	7%	No RT: 24% *p* = 0.009
Strom 2014, Cancer [25] (retrospective)	Moffitt	30 Gy/5 fx (51%), 50–68 Gy/25–36 fx (44%) (2–4 cm margin) 84% electrons, 13% photons, 7% other	Resected DM between 1989 and 2010	277 (164 obs, 113 postop RT)	Excision +/− RT	Median 43.1 months	7%	No RT: 17%
Chen 2008, Cancer [26] (retrospective)	Sydney Cancer Center	28.5–40 Gy/5–10 fx (22%), 48–50 Gy/20–25 fx (48%), 54–64 Gy/27–32 fx (30%) 33% orthovoltage, 36% MV ohotons, 30% electrons, 11% mixed	Resected DM between 1996 and 2007	128(101 obs, 27 postop RT)	Excision +/− RT	Median 40.5 months	7%	No RT: 6%
Foote 2008, ANZ J Surg [27] (retrospective)	Princess Alexandra Hospital	Mostly 48Gy/20 fx (3–4 cm margin) 63% electrons, 37% photons	DM between 1997 and 2006	24	Excision + RT	Mean 3 years		Relapsed disease 21% (*n* = 24)
Vongtama 2003, Head & Neck [28] (retrospective)	UCLA	44–66 Gy/22–33 fx (2 cm margin) Mostly electrons	Resected DM between 1976 to 1997	21 (7 obs, 14 postop RT)	RT	Mean 64.7 months	0%	No RT: 57%

Abbreviations: CI = confidence interval, DM = desmoplastic melanoma, fx = fractions, Gy = gray, HR = hazard ratio, LR = local recurrence, obs = observation, postop = post-operative R0 = margin-negative, RT = radiation therapy.

## Data Availability

No new data were created or analyzed in this study. Data sharing is not applicable to this article.

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
