# Peer review of "Adjuvant Radiation Therapy in Desmoplastic Melanoma: A Scoping Review"

_cancers, 2024, doi:10.3390/cancers16223874_

Round 1
Reviewer 1 Report
Comments and Suggestions for Authors
Report for the Authors:
Dear Authors,
I would like to congratulate you on a thorough and well-structured review on the role of adjuvant radiation therapy (RT) in the management of desmoplastic melanoma (DM). Below are some specific comments and suggestions to enhance your manuscript further:
Clarity and Readability:
The overall content is well-organized, but some sentences are overly complex, which might affect readability. Simplifying these would make the text easier to follow for a broader audience.
Quantitative Data Presentation:
The review provides valuable quantitative data supporting the use of adjuvant RT. However, the integration of these data points can be made more impactful by discussing potential meta-analytic approaches. For example, by pooling data from studies on local control and recurrence rates, you could provide a more robust quantitative estimate of the benefits of adjuvant RT. A formal meta-analysis or forest plot could enhance this discussion.
Discussion on Sentinel Lymph Node Biopsy (SLNB):
You note the low rate of sentinel lymph node (SLN) positivity in pure desmoplastic melanoma (pDM) patients. However, this point warrants a deeper exploration of potential explanations for this finding. Hypotheses such as the limited lymphatic involvement of pDM or its unique biological behavior could offer valuable insight for readers. Expanding this discussion would provide a more comprehensive understanding of why SLNB may not be as crucial in these cases.
Further Emphasis on Newer Therapies:
The integration of newer systemic therapies, such as immune checkpoint inhibitors, is excellent. However, it would be helpful to further emphasize how these therapies, in combination with RT, could alter treatment paradigms, especially in patients with advanced or unresectable disease. For example, recent findings on combined anti-PD-1 and RT therapies in melanoma could be discussed in greater depth.
Visual Summaries:
The inclusion of a PRISMA diagram is useful, but adding additional visual aids, such as a comparative table of outcomes from key studies, could improve the accessibility of the data. A table summarizing recurrence rates or local control rates from the studies you reviewed would make the comparisons more immediate for readers.
Future Research Directions:
The section on future research directions could be expanded. For instance, proposing specific research questions or trial designs, such as the combination of immunotherapy and RT in metastatic DM, could provide a more forward-looking conclusion. Additionally, further exploration of SLNB in pDM patients may offer valuable insights.
Overall, the review is a valuable contribution to the field, and with some refinements in language, structure, and data presentation, it will become an even more authoritative resource
Comments on the Quality of English Language
As previously commented, the quality of English in the manuscript is generally good, but there are certain areas where improvements in clarity, grammar, and sentence structure could make the text more fluid and easier to understand
Reviewer 2 Report
Comments and Suggestions for Authors
Having read the manuscript I have the following comments.
1. In this systematic review no specific information on the type and duration of RT used in the cited studies is given. The authors need to state what RT was used and was there a pattern observed in the studies they have discussed in this review. Please add this information as descriptions on the specific RT used in these cited studies is lacking.
2. P1 L32/3, replace "male sex" with "male gender".
P2 L58/65 you mention two subsets of DM, can you please describe these subsets in greater detail especially in regards their biological differences.
4. P2 L72, define SLNB, on L93 define SWOG.
5. Fig 1 and Table 1 are mentioned on P3 yet are shown on P5 and 6, please move to P3 or P4.
6. P4 L146, define LC
7. All references citations should be consistent, ie. either full journal names or abbreviated but not both, please correct.
Round 2
Reviewer 1 Report
Comments and Suggestions for Authors
Dear Authors,
I want to thank you for your detailed and thoughtful response to my comments. The simplification of language throughout the manuscript has improved readability, and the adjustments to Table 1, especially the clearer breakdown of local recurrence rates with and without RT, greatly enhance data accessibility. Your additional discussion on combination therapies, such as anti-PD1 with RT, provides valuable insights, and I understand your decision to keep the focus on adjuvant RT, leaving a more detailed discussion of SLNB for future studies. These changes have strengthened the manuscript significantly, and I appreciate the care you took in addressing each point.
Best regards,
Author Response
Comments: I want to thank you for your detailed and thoughtful response to my comments. The simplification of language throughout the manuscript has improved readability, and the adjustments to Table 1, especially the clearer breakdown of local recurrence rates with and without RT, greatly enhance data accessibility. Your additional discussion on combination therapies, such as anti-PD1 with RT, provides valuable insights, and I understand your decision to keep the focus on adjuvant RT, leaving a more detailed discussion of SLNB for future studies. These changes have strengthened the manuscript significantly, and I appreciate the care you took in addressing each point.
Response: Thank you for your detailed and constructive feedback. We appreciate your insights, especially regarding the language simplification, Table 1 adjustments, and expanded discussion on combination therapies. Your suggestions have significantly strengthened our manuscript.
Reviewer 2 Report
Comments and Suggestions for Authors
Having read the revised manuscript I have the following comments.
1. P5 Table 1 does not have a legend please add.
2. Ref 15 authors names should not be in capitals please correct.
3. Ref 26 the page number is e119716 as seen in PubMed. Please correct.
Author Response
Comment 1:P5 Table 1 does not have a legend please add.
Response 1: We have added a legend for table 1.
Comment 2: Ref 15 authors names should not be in capitals please correct.
Response 2: Ref 15 has been changed from capital to lowercase.
Comment 3: Ref 26 the page number is e119716 as seen in PubMed. Please correct.
Response 3: The correct page number has been added to the reference